# Running Performance during the Holy Month of Ramadan in Elite Professional Adult Soccer Players in Russia

**DOI:** 10.3390/ijerph182111731

**Published:** 2021-11-08

**Authors:** Eduard Bezuglov, Oleg Talibov, Vladimir Khaitin, Bekzhan Pirmakhanov, Zbigniew Waśkiewicz, Mikhail Butovskiy, Ryland Morgans

**Affiliations:** 1Department of Sports Medicine and Medical Rehabilitation, Sechenov First Moscow State Medical University of the Ministry of Health of the Russia Federation, 119991 Moscow, Russia; e.n.bezuglov@gmail.com (E.B.); z.waskiewicz@awf.katowice.pl (Z.W.); Rylandmorgans@me.com (R.M.); 2“Smart Recovery” Sports Medicine Clinic LLC, 121552 Moscow, Russia; drmike81@inbox.ru; 3PFC CSKA, 125252 Moscow, Russia; 4Russian Football Union, 115172 Moscow, Russia; 5High Performance Sport Laboratory, Moscow Witte University, 300028 Moscow, Russia; oleg.talibov@gmail.com; 6Sirius University of Science and Technology, 354349 Sochi, Russia; 7Department of Internal Medicine, Clinical Pharmacology and Emergency Medicine, Moscow State University of Medicine and Dentistry, 127473 Moscow, Russia; 8Department of Physical Methods of Treatment and Sports Medicine, Pavlov First Saint Petersburg State Medical University, 197022 St. Petersburg, Russia; khaitinvladimir@gmail.com; 9FC Zenit, 197341 St. Petersburg, Russia; 10Department of Epidemiology, Biostatistics and Evidence-Based Medicine, Faculty of Medicine and Health Care, Al-Farabi Kazakh National University, Almaty 050040, Kazakhstan; 11FC Kairat, Almaty 050054, Kazakhstan; 12Institute of Sport Science, Jerzy Kukuczka Academy of Physical Education, 40-065 Katowice, Poland; 13FC Rubin, 420036 Kazan, Russia

**Keywords:** Ramadan, soccer players, soccer, match performance

## Abstract

Religious fasting in the Holy Month of Ramadan is an important element of the Muslim culture during which no eating or drinking is permitted from dawn till dusk. A considerable number of Muslim soccer players abide by these restrictions, which may cause a negative impact on key running performance parameters during competitive matches. Alterations to diet and water intake during the Holy Month of Ramadan may affect various running performance parameters in elite Muslim professional adult soccer players. This study was conducted with two groups of soccer players from the Russian Premier League (RPL): The Exposure Group (EG) consisted of 13 Muslims age 24.0 ± 2.8 years abiding by religious fasting and the Control Group (CG) included 13 non-Muslim age 26.0 ± 4.4 years. Using the Instat system, the running performance of each player was controlled in both groups during matches from the RPL before and in the third week of Ramadan (a total of two matches for every player). None of the measured parameters demonstrated significant changes in any match. In conclusion, restrictions in diet and liquid intake during the Holy Month of Ramadan had no negative influence on the running performance of elite Muslim professional adult soccer players during daytime matches.

## 1. Introduction

Soccer is one of the most popular sports in the world [1] played by many people including Muslims [2]. With the ever-evolving rise in the general population and the popularity of soccer, it is estimated that over 131 million players from Asia and Africa, two regions most populated by Muslims, are competing and ~25% of the world population will be practicing Islamic religion by 2022 [3,4]. Therefore, with the growing popularity of soccer in Islamic countries it can be purported that sport professionals need to better understand Muslim soccer players, especially during Ramadan. Ramadan is a Holy Month of obligatory fasting, during which faithful Muslims refrain from food and liquids during daylight [2]. Food is consumed before sunrise prior to the Morning Prayer and after sunset following the Evening Prayer [2]. Thus, the duration of fasting during the day depends on the number of daylight hours, which also changes every year due to the month of Ramadan being determined by the lunar calendar [2].

Since the majority of Muslim soccer players continue to train and compete during the Holy Month of Ramadan, sport professionals need to be fully informed regarding the potential influence on physical performance [5,6]. Existing evidence on the effects of Ramadan on general health varies. Ramadan fasting is considered safe for healthy people, but those with various medical conditions are advised to consult their physicians [7,8]. 

Several studies have highlighted an adverse effect of Ramadan fasting on exercise performance (Aziz and Png, 2008; Chaouachi et al., 2009; Waterhouse, 2010) [9,10,11], although, Shephard et al. found that the effects of Ramadan on athletic performance are insignificant if training is optimized, and sleeping habits, diet and fluid intake are consistent [12]. However, the influence of Ramadan on the physical match performance in soccer has previously been reported [2,6,7], although there is only limited information and conflicting conclusions regarding this concept [2,6,7]. According to Stolen et al., in professional soccer match-play the total distance covered by outfield players was ~10–12 km, at an average work intensity of 80–90% of HRmax. Approximately 1–11% of this total distance involved sprinting, with an average sprint of 2–4 s every 90 s. There were also 1000–1400 cyclical short activities changing every 4–6 s including high intensity running every 70 s, approximately 15 tackles, 10 headers, 50 involvements with the ball, 30 passes of varying distances, and dueling activities [13]. Thus, soccer match-play can be considered a physically demanding sport where appropriate nutritional preparation is required.

However, none of the existing research involved soccer players during competitive match-play from a top division. Currently, the analysis of match performance employing sophisticated tracking systems is common practice in professional soccer [14]. Despite the market variety, all existing systems use video-based multi-tracking to translate registered information into numerical values for further calculation and estimation of running performance [14], although none of the preferred methods are standardized for running performance analysis. Thus, the most widespread method involves mathematical modelling of data from fixed cameras located around the stadium at roof height [15]. The output data consists of a generated report on the running performance of all players, including the following metrics: total distance; average speed; distance covered within the different speed thresholds (high-intensity and sprinting), number of accelerations and decelerations; and maximal speed. The specific metric thresholds may vary depending on the individual tracking system [15].

Therefore, due to scant and inconsistent literature examining the influence of Ramadan on running performance in professional soccer, this study aimed to evaluate the relationship between Ramadan fasting and running performance during competitive match-play of Muslim soccer players competing in the Russian Premier League (RPL).

Our hypothesis was that fasting during daytime and restricted water intake during the holy month of Ramadan may affect various measures of motor activity (such as total distance, high-intensity distance and sprint distance) during competitive matches in elite Muslim professional adult soccer players.

## 2. Methods

### 2.1. Ethical Approval

The study was approved by the local Ethical Committee of the Sechenov First Moscow State Medical University with the number N 11-19. All subjects gave their written informed consent for the analysis of their running performance during competitive matches.

### 2.2. Subjects

All subjects in the Exposure Group (EG) met the following inclusion criteria: age > 18 years, strict fasting during the Holy Month of Ramadan for the past five years, no injuries during the analyzed period, a registered player for a RPL club for at least one year before the study period. Those players who did not meet at least one of the inclusion criteria were excluded. The study included two competitive matches played in the RPL at official stadiums where Instat match tracking systems were installed. Matches were scheduled to finish before sunset, i.e., the time when fasting players were allowed drink and food intake. Matches not meeting these criteria were excluded. Goalkeepers and injured players were also excluded from the study.

The study was comprised of 26 players of the RPL. The EG consisted of 13 Sunni Muslims (age 24.0 ± 2.8 years), fully adherent to the restrictions of Ramadan, neither eating nor drinking during daylight hours (from about 5.00 a.m. till 9.00 p.m.). All players from the RPL that fulfilled the study criteria were included. The Control Group (CG) was composed of 13 non-Muslim players (age 26.0 ± 4.4 years) from the same clubs. Players in the CG were chosen as matched pairs. Therefore, from the same club, same position and had similar age and BMI. The demographic data were collected from team physicians. All Muslims had fasted during the previous five years and believed that it did not restrict their physical activity.

Training processes and the measures of warm-up before each match did not change during the Holy Month of Ramadan in the EG subjects. All subjects played in regular matches of the RPL on days immediately before Ramadan (between the 7 and the 20 April 2019) and during the third week of Ramadan (from the 19 till the 26 May 2019). All matches took place during daylight hours, starting between 2.00 p.m. and 7.00 p.m. The outside temperature ranged from 9 to 16 °C during the matches before Ramadan, and from 16 to 26 °C during matches at the end of Ramadan. Thus, prior to every match each fasting player did not consume any food or liquid for 9–14 h and the EG subjects also did not consume any items at half-time. All subjects played in the same position during both matches and maintained an injury-free status. Subjects were unaware that the influence of Ramadan was being examined against match running performance. The characteristics of subjects are presented in Table 1.

### 2.3. InStat Kinematic System

Currently, various video-based systems track performance indicators of soccer players (InStat, Optasport, Wyscout). Such platforms quickly and accurately provide a large range of match-related performance measures, allowing the simultaneous analysis of the physical efforts, movement patterns, and technical actions of players, both with and without the ball [16]. The match performance indicators for each player were determined by the position-specific InStat system. The InStat tracking system has previously been employed to analyze the association between running performance and game performance indicators in professional soccer players [16].

The InStat kinematic system captured the outfield players using six cameras placed around the perimeter of the field at the minimal height of 12 m. The frame frequency was 25 frames per second where data was centralized for further analysis. The following parameters of running performance were selected to estimate the match performance of players: total distance covered per match and during each half (m), the average speed per match and during each half (km/h), maximal speed (km/h); the total distance covered at high-intensity (m) (speed range 19.8–25.2 km/h) per match and for each half, the total distance covered sprinting (m) (speed above 25.2 km/h) per match and for each half, and the number of sprints. The speed thresholds for each category are similar to those reported previously [16] and have been universally accepted.

### 2.4. Statistical Analysis

Statistical analysis was conducted using the GraphPad Prism software 8.0.0 version for Mac OS X. No imputation or substitution of missing values was performed. Normality of the quantitative data was tested using the Kolmogorov-Smirnov test. Normally distributed data were described using mean (M), standard deviation (SD), and min-max ranges. For other distributions median (Me), interquartile intervals (Q1–Q3), and min-max ranges were used. A two-sample independent *T*-test with Welch’s correction for unequal variances was used to assess the intergroup differences (age, height, weight and BMI) in case of normal distribution. The Mann-Whitney U-test was used to assess the significance of intergroup differences for playing minutes distributed non-normally. For the primary outcomes (functional loading parameters), the Wilcoxon test for paired measures was used as a more conservative and robust test for the relatively small number of observations. Difference of performance measures in CG and EG were also tested with non-parametric test (Mann-Whiney U-test). Categorical data (player position) were described using frequency charts showing an absolute value and its percentage share. Chi-squared test was used to estimate the differences of player position in 3 × 2 contingency tables. Values at *p* < 0.05 were considered statistically significant.

## 3. Results

No statistical differences were found between the EG and CG. The comparison of playing minutes in matches before and during Ramadan showed no statistical differences (Table 2).

Referring to the data presented in Table 2, no statistical difference in physical match performance was seen between the matches completed prior to Ramadan and during the end of the third week of Ramadan. Those players who adhered to food and water restrictions demonstrated no decrease in functional performance. There was no statistical difference in functional performance between the EG and CG. The results obtained prior to Ramadan and during week 3, *p* > 0.05 was observed for all comparisons. The comparison of the main performance measures in the 1st and 2nd half is presented in Table 3. Such factors as average speed in the EG and total distance in the CG had significant difference—these results were higher in matches performed in the third week during Ramadan. The comparison between all measures of both the EG and the CG before and during week three showed no statistically significant differences (*p* > 0.05).

## 4. Discussion

This study aimed to investigate the relationship between religious fasting during Ramadan and running performance in elite Muslim soccer players competing in the RPL. The first notable finding was that, while abiding by strict, religious fasting, running performance of elite Muslim professional adult soccer players in competitive matches showed no changes during the period of Ramadan. These results can possibly be explained by other external factors (i.e., tournament situation in the last match of the championship). The second significant finding was that the decrease in running performance during the second half of matches for fasting players was comparable with non-fasting players. This decrease was not larger in matches completed during the third week of Ramadan. 

These results support the work conducted by Carling et al. investigating match performance in a single competitive match of the 3rd division in the French Championship on the third day of the Holy Month of Ramadan [17]. However, it should be noted that the running performance analyzed in this research was from one match at the beginning of the religious fasting (on the third day of Ramadan). Furthermore, in the study by Carling et al. the matches took place in the evening and players had an opportunity to consume isotonic beverages during the half-time break. Additionally, Carling et al. indicated that the Muslim players examined may save energy in the 1st half of the match as they were aware that the Ramadan protocol followed may have a negative impact of their physical abilities during the match. Furthermore, the small sample size in the study should be considered (i.e., seven Muslim soccer players with only four playing the full match, and five non-Muslim players included in the CG) [17]. Furthermore, Kirkendall et al. observed no negative impact on objective measures of physical performance, such as sprinting and vertical jumps in young Muslim soccer players [18]. In contrast, Aziz et al. reported that the influence of Ramadan on the running performance of young amateur soccer players during a series of matches showed a deteriorating effect on various markers of running performance during the second half of the match [19]. 

Aziz et al. analyzed 13 young soccer players and revealed a negative effect of fasting during Ramadan on different parameters of running performance in the group of fasting players [19]. The running performance was analyzed a week before fasting and at the end of the third week of the Holy Month of Ramadan. The majority of parameters (total distance, high-intensity distance) declined in comparison with the CG. Notably, the subjects were young amateur soccer players and the use of structured recovery modalities post-match and post-training were unlikely. Furthermore, the matches were performed between the players of one club team making it less competitive [19].

Zerguini et al., 2007 and Zerguini et al., 2012 explored the effect of Ramadan on various parameters of physical performance in soccer players, where the authors reported no significant negative contribution [2,5]. However, those studies were conducted on non-elite players and in non-competitive conditions [2,5]. It should also be noted that data on the positive effects of timely ingestion of fluids, carbohydrates, proteins and fats on the recovery processes after physical activity have been obtained [20,21,22,23]. Mimiran et al. reported that Ramadan fasting may be accompanied by a moderate improvement of lipid and lipoprotein parameters [24], while Alabbood et al. stated that in their review most studies found an improvement or no change in glycemic control parameters during Ramadan fasting [25].

However, in our study, the timely intake of fluids and nutrients could not be controlled, and no negative effect was reported due to their absence, at least not evidently in the running performance during match-play. Possibly, it is necessary to consider that not all athletes maintain their strict diet on the day of competition. The study conducted by Farooq et al. examined three national soccer teams, with a majority of Muslim players, competing in the Olympic Games held in London 2012. Findings reported that only 39% of interviewed soccer players intended to fast during the competition. In addition, none of 54 players planned to fast on the day of the match [26]. Thus, it can be assumed that in a short period of time the lack of fluid and food intake during and post-match-play can be compensated by the internal reserves of the body. The strength of our study is that, according to our data, for the first time we studied the relationship between fasting during Ramadan and running performance during regular competitive match-play in the RPL. 

Our study also highlighted a number of limitations. The lack of running performance data after the end of Ramadan can be considered a limitation. Moreover, the dynamic changes of hematological parameters, cardiovascular function, and subjective fatigue after matches were not measured, although this would allow an evaluation of the actual influence of Ramadan on the entire organism. Several studies have reported no significant disturbances in sleep architecture between fasting and non-fasting non-athlete populations [27], while Rocky et al. further noted that Ramadan decreased nocturnal sleep duration and sleep time was delayed, which may have negative effects on daytime performance [28]. These concepts are noteworthy in professional athlete populations. 

Future research should focus on the impact of Ramadan on different age group athletes, previously accepted field-based measures of physical performance [29], and the epidemiological analysis of injuries in Muslim players, such as tendinopathies [30]. Furthermore, the holistic estimation of the dynamic changes in various psycho-physiological parameters before, during, and after the Holy Month of Ramadan would also be of interest.

## 5. Conclusions

Restrictions to diet and liquid intake during the Holy Month of Ramadan had no adverse effect on match performance in two competitive matches among adult professional soccer players from the RPL. 

## Figures and Tables

**Table 1 ijerph-18-11731-t001:** Subject characteristics.

	Exposure Group (*n* = 13)	Control Group (*n* = 13)	*p*-Value
Age (M ± SD; min–max)	24.0 ± 2.8; 20–30	26.0 ± 4.4; 19–32	0.4058
Body Height (M ± SD; min–max)	181.0 ± 5.8; 173–190	184.0 ± 9.3; 170–196	0.2468
Body Weight (M ± SD; min–max)	74.0 ± 5.0; 61–81	78.0 ± 8.5; 65–91	0.2294
BMI (M ± SD; min–max)	23.0 ± 0.93; 21–25	23.0 ± 1.2; 21–26	0.8711
Playing Minutes before Ramadan (Me; Q1–Q3; min–max)	91; 87–95; 73–96 ^#^	94; 90–96; 90–98 ^##^	0.2513
Playing Minutes third week of Ramadan (Me; Q1–Q3; min–max)	95; 92–95; 73–98 ^#^	95; 92–95; 92–98 ^##^	0.5777
Player Position			
Defender; *n* (%)	3(23)	4(31)	0.6420
Midfielder; *n* (%)	6(46)	7(54)	
Forward; *n* (%)	4(31)	2(15)	

^#^—*p* = 0.1548 (within-group Wilcoxon test) ^##^—*p* = 0.2080 (within-group Wilcoxon test).

**Table 2 ijerph-18-11731-t002:** Physical match performance (M ± SD).

Parameters	Exposure Group		*p*	Control Group		*p*
	Before	Week 3		Before	Week 3	
Total Distance (m)	10,613 ± 1059	10,654 ± 935.2	0.8926	10,643 ± 1041	10,952 ± 1078	0.3054
Average Speed (km/h)	3.6 ± 0.54	3.6 ± 0.61	0.1606	3.6 ± 0.58	3.6 ± 0.65	0.7993
High Intensity Distance (m)	760 ± 214	863 ± 198	0.0942	632 ± 268	778 ± 275	0.1099
Sprint Distance (m)	130 ± 112	189 ± 61	0.0681	108 ± 71	118 ± 90	0.5195
High intensity and sprint count (n)	62 ± 14.5	71.5 ± 14	0.0942	59 ± 23	66 ± 23	0.1514
Maximal Speed (km/h)	29.9 ± 2.38	30.2 ± 1.3	0.5317	28.4 ± 1.98	29.2 ± 1.26	0.2241

**Table 3 ijerph-18-11731-t003:** Physical match performance per half, 1st half versus 2nd half (M ± SD).

Parameters	Exposure Group		*p*	Control Group		*p*
	Before	Week 3		Before	Week 3	
Total Distance (m)	0.99 ± 0.31	0.94 ± 0.9054	0.1094	0.93 ± 0.068	1.0 ± 0.049	0.0012 ^^
Average Speed (km/h)	0.94 ± 0.54	0.96 ± 0.041	0.0288 ^	0.90 ± 0.098	0.94 ± 0.054	0.1973
High Intensity Distance (m)	0.99 ± 0.42	1.0 ± 0.93	0.8523	0.99 ± 0.29	0.90 ± 0.22	0.4548

^—median difference 0.03 (95%CI 0.01–0.06) ^^—median difference 0.08 (95% CI 0.01–0.11).

## Data Availability

The raw data supporting the conclusions of this article will be made available by the authors, without undue reservation.

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
