# Peer review of "Running Performance during the Holy Month of Ramadan in Elite Professional Adult Soccer Players in Russia"

_ijerph, 2021, doi:10.3390/ijerph182111731_

Round 1
Reviewer 1 Report
This is a very interesting paper about differences in sport performance between Muslim and non-Muslim soccer players. Topic can be interested to the reader since Muslim players are a big group.
The article is clear and well written and methods are fine. Tables are clear and very useful. Great job!
I have only two suggestions:
1) I think that some sentences about the existing literature and papers similar to your could be moved from the introduction to the discussion section. Try to be more coincise in the first part of the paper. The discussion can be improved with what you wrote before.
2) I think that future suggestions could be linked to other filed test previously tested in soccer (https://www.mdpi.com/2411-5142/6/3/59) and to epidemiological studies about injury in musilm players, such as tendinopathies (http://rua.ua.es/dspace/handle/10045/108922). I ask you to consider these two references
Author Response
Reviewer 1
Dear Reviewer,
Thank you very much for your detailed reviewing our paper. We know that finally it increases its quality and we are very happy that you spend so much time to help us to improve it. All the changes we mark with red color in the text.
Authors
I have only two suggestions:
- I think that some sentences about the existing literature and papers similar to your could be moved from the introduction to the discussion section. Try to be more concise in the first part of the paper. The discussion can be improved with what you wrote before.
Thank you for you specific and professional suggestion. We agree with this and we moved some parts from introduction section to discussion. Now text looks much better and we hope that it fulfils your expectations.
Moved to Discussion Line 223
Mimiran et al. reported that Ramadan fasting may be accompanied by a moderate improvement of lipid and lipoprotein parameters [11].
Moved to Discussion Line 225
While Alabbood et al. stated that in their review most studies found an improvement or no change in glycemic control parameters during Ramadan fasting [12].
Moved to Discussion Line 231
Possibly, it is necessary to consider that not all athletes maintain their strict diet on the day of competition. The study conducted by Farooq et al. examined three National soccer teams, with a majority of Muslim players, competing in the Olympic Games held in London 2012. Findings reported that only 39% of interviewed soccer players intended to fast during the competition. In addition, none of 54 players planned to fast on the day of the match [13].
Moved to Discussion Line 246
Several studies have reported no significant disturbances in sleep architecture between fasting and non-fasting non-athlete populations [10].
Moved to Discussion Line 248
Rocky et al. further stated that Ramadan decreased nocturnal sleep duration and sleep time was delayed, which may have negative effects on daytime performance [18].
Moved to Discussion Line 204
Kirkendall et al. observed no impact on objective measures of physical performance, such as sprinting and vertical jumps [19].
Moved to Discussion Line 206
Aziz et al. also reported the influence of Ramadan on the running performance of young amateur soccer players during a series of matches and showed a deteriorating effect on various markers of running performance during the second half of the match [21].
- I think that future suggestions could be linked to other filed test previously tested in soccer (https://www.mdpi.com/2411-5142/6/3/59) and to epidemiological studies about injury in Muslim players, such as tendinopathies (http://rua.ua.es/dspace/handle/10045/108922). I ask you to consider these two references
Both references have been cited in text and inserted in Reference list [33, 34].

Reviewer 2 Report
The data presented and discussed here is interesting, as it analyses the relationship between Ramadan fasting and running performance during competitive matches of Muslim soccer players competing in the elite leagues of European soccer. As the main findings, the authors found that the Holy Month of Ramadan had no adverse effect on match performance in two competitive matches among adult professional soccer players from the Russian Premier League.
Despite the interesting work, I strongly suggest following the comments to improve the quality of the manuscript. The manuscript needs some written reviewing.
Abstract:
1. It is not clear what the main objectives, methods, results, and conclusions of the current work are. Please review all these sections, and be more specific.
- I suggest adding some characteristics of your sample (e.g., age);
- Please present all data needed (include 95% IC of all [i.e, main] results).
Subject and methods:
2. Subjects and methods? I suggest changing only for "Methods" or "Materials and methods".
3. Subjects. I think there is a typo here: "age > 13", should be this "age >18"?
- It is not necessary to repeat age in the exclusion criterion if it was already mentioned in the inclusion criterion. Please review and confirm.
- In fact, review all inclusion and exclusion criterion, because seems that all sentences are repeated.
4. Line 128. "sunset". Please be more specific.
5. Authors should be more clear about how each group was selected, ie, how was really made the allocation of each participant for each group. It was randomly? blinded? etc... Please, be more specific.
6. I suggest adding a flowchart to better follow the study design and procedures.
7. Statistical procedures might need to be discussed using a within-subjects approach since basic group comparisons were performed.
- A two-sample independent T-test, Mann-Whitney U-test, Wilcoxon test for paired measures were used to compare the measured variables. How was this comparison attempted? Did authors pool data for the comparison? How many data points were paired?
- Given the intra-individual variability, a within-subjects approach (recommended for small samples) might be appropriate (please see some recent works: https://pubmed.ncbi.nlm.nih.gov/31527865/, https://pubmed.ncbi.nlm.nih.gov/33672683/; https://www.frontiersin.org/articles/10.3389/fphys.2021.678462/full).
- I suggest analyzing your data for each player as individually.
- I recommend presenting some individual figures, showing the main outcomes/results across the observation period.
- I suggest including 95% CI in your results.
Despite the same "competitive level" (i.e., elite), for each group, as mention by the authors, it is hard to understand why and how was this comparison really attempted? Once again, I truly recommend analyzing your data using a within-subjects approach to better understand your data.
Author Response
Dear Reviewer,
Thank you very much for your detailed reviewing our paper. We know that finally it increases its quality and we are very happy that you spend so much time to help us to improve it. All the changes we mark with blue color in the text. We hope that some of the Reviewers’ suggestions which treat are disputable and did not include in the revision will find his understanding.
Authors
The data presented and discussed here is interesting, as it analyses the relationship between Ramadan fasting and running performance during competitive matches of Muslim soccer players competing in the elite leagues of European soccer. As the main findings, the authors found that the Holy Month of Ramadan had no adverse effect on match performance in two competitive matches among adult professional soccer players from the Russian Premier League.
Despite the interesting work, I strongly suggest following the comments to improve the quality of the manuscript. The manuscript needs some written reviewing.
Abstract:
1. It is not clear what the main objectives, methods, results, and conclusions of the current work are. Please review all these sections, and be more specific.
- I suggest adding some characteristics of your sample (e.g., age);
Sample size and players age are added
- Please present all data needed (include 95% IC of all [i.e, main] results).
We suppose that the main result was absence of significant changes. The data and exact p values are provided in the text but couldn’t be reproduced in the Abstract in compact way.
Subject and methods:
- Subjects and methods? I suggest changing only for "Methods" or "Materials and methods".
Changed to “Methods”
- Subjects. I think there is a typo here: "age > 13", should be this "age >18"?
Corrected
- It is not necessary to repeat age in the exclusion criterion if it was already mentioned in the inclusion criterion. Please review and confirm.
Corrected
- In fact, review all inclusion and exclusion criterion, because seems that all sentences are repeated.
Corrected
- Line 128. "sunset". Please be more specific.
That means the time when drinking and eating is allowed. Added to the text.
- Authors should be more clear about how each group was selected, ie, how was really made the allocation of each participant for each group. It was randomly? blinded? etc... Please, be more specific.
In the exposure group included all players who fulfilled in/excl criteria. Control group was composed from matched pairs, chosen from the same clubs. Details are added.
- I suggest adding a flowchart to better follow the study design and procedures.
Since the study was observational and no any specific procedures were planned the flow-chart was not used.
- Statistical procedures might need to be discussed using a within-subjects approach since basic group comparisons were performed.
- A two-sample independent T-test, Mann-Whitney U-test, Wilcoxon test for paired measures were used to compare the measured variables. How was this comparison attempted? Did authors pool data for the comparison? How many data points were paired?
There are no pooled data. Subjects in the groups were compared before fasting time (match 1) and 3 weeks after (match 2). Their performance was compared with non-fasting players who took participation in the same games.
- Given the intra-individual variability, a within-subjects approach (recommended for small samples) might be appropriate (please see some recent works: https://pubmed.ncbi.nlm.nih.gov/31527865/, https://pubmed.ncbi.nlm.nih.gov/33672683/; https://www.frontiersin.org/articles/10.3389/fphys.2021.678462/full).
- I suggest analyzing your data for each player as individually.
- I recommend presenting some individual figures, showing the main outcomes/results across the observation period.
Initially the study was planned as a pilot retrospective study on the limited sample with only one comparison within groups. As we understand in this case within-subject calculations couldn’t give reliable data. Anyway this suggestion is very important and intra-individual variability will be used in the upcoming prospective cohort study when performance will be analyzed in several matches during the Fast.
- I suggest including 95% CI in your results.
95% CI are given in the table 3 for the comparisons with p<.05
Despite the same "competitive level" (i.e., elite), for each group, as mention by the authors, it is hard to understand why and how was this comparison really attempted? Once again, I truly recommend analyzing your data using a within-subjects approach to better understand your data.
We are afraid that we do not understand fully this comment. The number of Muslim athletes at the competitive level who fast according to religious restrictions. We planned a retrospective study and wanted to describe some phenomenon. Maybe the adding “A retrospective study’ to the paper title would heal some methodological doubts?

Reviewer 3 Report
In my opinion it's a interesting subject, but the size of the sample it's to small to have same conclusions. Sorry
Author Response
Dear Reviewer,
Thank you very much for your reviewing our paper. However, the opinion is very scarce and focusing only on number of athletes we try to answer your comment. Generally, we have to say, that the number of Muslim athletes at the competitive level who fast according to religious restrictions. From the other side the number of participants in any research so problematic that we may show many arguments that our quantity of athletes is sufficient. We planned a retrospective study and wanted to describe some phenomenon. Maybe the adding “A retrospective study’ to the paper title would heal your methodological doubts?
Authors

Round 2
Reviewer 2 Report
I am happy with the current version of the manuscript.
The authors did a good job on reviewing the manuscript.
Reviewer 3 Report
Thanks for your effort with this papper.
I don´t think that you understund my major issue, it's not the number of players, that ok it's, but the number of games "The study included two competitive matches played in the RPL at official stadiums where Instat match tracking systems were installed."
With just 2 games it's very hard to get some conclusions, because we know that the quality of the opponent, the match status, etc can have influence in the parameters of running performance selected. In my opinion you could use a more individual aproach, looking for the individual diferences and not the group diferences.